# Metagenomics Provides a Deeper Assessment of the Diversity of Bacterial Communities in Polar Soils Than Metabarcoding

**DOI:** 10.3390/genes14040812

**Published:** 2023-03-28

**Authors:** Burkhard Becker, Ekaterina Pushkareva

**Affiliations:** Institute for Plant Sciences, University of Cologne, 50923 Cologne, Germany

**Keywords:** metagenomics, metabarcoding, polar regions, terrestrial communities, microbiomes

## Abstract

The diversity of soil bacteria was analyzed via metabarcoding and metagenomic approaches using DNA samples isolated from the biocrusts of 12 different Arctic and Antarctic sites. For the metabarcoding approach, the V3-4 region of the 16S rRNA was targeted. Our results showed that nearly all operational taxonomic units (OTUs = taxa) found in metabarcoding analyses were recovered in metagenomic analyses. In contrast, metagenomics identified a large number of additional OTUs absent in metabarcoding analyses. In addition, we found huge differences in the abundance of OTUs between the two methods. The reasons for these differences seem to be (1) the higher sequencing depth in metagenomics studies, which allows the detection of low-abundance community members in metagenomics, and (2) bias of primer pairs used to amplify the targeted sequence in metabarcoding, which can change the community composition dramatically even at the lower taxonomic levels. We strongly recommend using only metagenomic approaches when establishing the taxonomic profiles of whole biological communities.

## 1. Introduction

The ecosystems of the polar regions belong to the most extreme environments on Earth. While they are only sparsely inhibited by humans, they represent a fifth of our planet and comprise the majority of the world’s cryosphere. The polar regions still represent vast white wildernesses. However, even these remote regions are under anthropogenic influence, with climate change significantly altering these icy environments in the near future. Current estimates predict these cold regions to decrease by at least 16% by 2099 according to the Global Change Assessment Model RCP4.5 [1]. In addition, freeze/thaw cycles will be more common in huge areas of the polar regions [2]. These will result in massive changes to biodiversity and ecosystem functioning [3].

While the macrofauna and -flora (plants and algae) of the polar regions have been studied quite extensively, we know much less about microbial organisms and communities. This is especially the case for the terrestrial ecosystems, which are nearly completely (Antarctica) or mainly (Arctic) dominated by lichens [4,5] and biological soil crust communities [6] and will, due to climate change, undergo dramatic changes. Historically, identification of members of these communities required biochemical (bacteria) or microscopical (cyanobacteria, micro algae, micro fauna) approaches, often after establishing cultures for the organisms of interest [7]. However, it has been known since the middle of the 1980s that only a small percentage (estimated to be 0.1 to 5%) can be cultured [7]. With the advent of DNA-sequencing techniques, identification was more commonly based on sequence information of some marker molecules (most often 16S rRNA for bacteria and 18S rRNA for eukaryote sequences) isolated from cultured material. More recently, metabarcoding has been introduced to establish the composition of complete microbial communities [8,9]. Metabarcoding requires the isolation of DNA from environmental samples followed by the amplification (by PCR) of a specific biomarker (often certain variable regions of the rRNA) which serves as a barcode for the identification of the presence of an organism. The frequency with which an RNA sequence is found in a sample is a measure of the abundance of an RNA sequence. Individual RNA sequences are commonly referred to as amplicon sequence variant (ASV) or operational taxonomical unit (OTU) and represent a species. Metabarcoding has been widely used in the last years to study community composition of soil samples (e.g., [9,10,11]).

A major problem with metabarcoding is the PCR step [12,13,14]. Universal primer pairs targeting all organisms have not been found so far. Therefore, often more specific primer pairs, which target only some organismal groups (bacteria [15] and eukaryotes [16] but also specific primers for lower taxonomic groups e.g., Cercozoa [17], Xantophyceae [18]) are used. This results in difficulties in establishing taxonomic profiles of whole communities.

To overcome this problem, metagenomic approaches have been proposed. Metagenomics is the random sequencing of DNA isolated from environmental samples (e.g., soil), thus without a PCR step. Over the last years sophisticated bioinformatic tools were developed to establish, in principle, most if not all genomes present in microbial communities. This huge sequence information can be used to investigate the biodiversity but also to analyze the functions of individual members in the community (see [19] for a comprehensive discussion of metabarcoding and metagenomics in soil science). Currently, the success of this approach seems only to be impeded by sequencing depth and computer speed and time.

However, a real comparison of metabarcoding and metagenomics are rare [20,21]. We therefore decided to use the same DNA samples isolated from Svalbard (the Arctic) and East Antarctica for a comparison of metabarcoding and metagenomic approaches using the same bioinformatic pipeline. We will not discuss any ecological aspects of the results, which have been already published for the Arctic samples [11] and will be published soon for the Antarctic samples (Pushkareva et al., submitted).

## 2. Material and Methods

### 2.1. Sampling Sites

The sampling sites for the Arctic and Antarctic samples are described in [11] and Pushkareva et al. (submitted), respectively. In brief, the Arctic samples S1, S3, S7, and S8 were collected in South Svalbard, while S11 was taken in the Longyearbyen (Svalbard) surroundings and NA3 near Ny-Alesund (Svalbard). Antarctic sample Pad2 was collected on Padda Island, and Amu8 and Amu14 were sampled at the Amundsen Bay in East Antarctica. Lang_37 and Skar18 were collected at Langhovde Hills and Skarvsnes Foreland, respectively, both located at Lützow-Holm Bay, East Antarctica. Syo6 was sampled close to Syowa Station at Ongul Island, East Antarctica. All samples are different types of biological soil crusts.

### 2.2. DNA Isolation, Sequencing, and Bioinformatic Analyses

DNA was isolated as described in [11]. Metabarcoding sequencing of the V3-4 region of rDNA of 4 replicates was performed as described in [11]. The raw reads for the Arctic and Antarctic samples are available at the Sequence Read Archive (SRA) under projects PRJNA881983 and PRJNA936101, respectively. For metagenomics, one replicate of DNA samples used for metabarcoding (replicate 1 of Pad2 and S1 sites, replicate 2 of S11 site and replicate 3 of Amu14 site) was sequenced by Eurofins (Ebersberg, Germany) and Biomarker Technologies (Münster, Germany). Additional replicates were sequenced only by Eurofins (replicate 1 for the Lang_37, NA3_R2, S3c, S7c, S8, and Skar18 sites; replicate 3 for the Amu8 site; and replicate 4 for the Syo6 site). The raw reads were submitted to SRA under project PRJNA945601 for the Eurofins data set and PRJNA948668 for the Biomarker data set.

Raw reads were demultiplexed and primers and poor sequences removed using the trimmomatic version (standard settings) [22] in the Omicsbox software package [23]. Taxonomic classification of the reads obtained via both metabarcoding (16S rDNA gene targeted) and metagenomic sequencing was performed using the Kraken 2 software [24] included in Omicsbox with standard settings. Briefly, Kraken 2 assigns taxonomic labels to DNA short reads by examining the k-mers within a read and querying a database with information on those k-mers for different species (see [24]). The current Kraken database 2022_8 contains information of 23862 genomes (mainly for bacteria (49%) and viruses (48%)). Differential abundance analyses to identify OTUs that differ significantly between two or more samples were performed using edgeR [25] using the standard settings as provided in Omicsbox.

## 3. Results and Discussion

DNA was isolated for six Arctic and six Antarctic sites (see Material and Methods for details). The same DNA samples were used for metagenomic and metabarcoding approaches. For the metabarcoding, the 16S rRNA gene was targeted. All reads were analyzed with the same software, Kraken 2, as implemented in the Omicsbox software using standard settings. Appendix A gives an overview about the number of reads classified, the estimated diversity within the samples using the Shannon and Simpson indices, and the percentage of classified reads at various systematic levels.

The percentage of classified reads was much higher for metabarcoding than for the metagenomic analyses. Kraken 2 classified 17–32% of all reads (average 22% ± 4%, n = 16) in analyses of the metagenomic data sets. In contrast, in the metabarcoding data sets between 77–97% of all reads (average 90% ± 11%, n = 36) were taxonomically classified. The higher classification rate within the metabarcoding data set was probably due to the targeted rDNA sequence in metabarcoding, as we attempted to amplify only the 16S rDNA. Generally, between 77% and 97% of the reads could be classified in metabarcoding. Only for three replicates was a lower classification value than 80% obtained (Appendix A). In contrast, using the same software, only 17–25% of the reads in the metagenomics dataset could be classified. This probably reflects the Kraken 2 database used for classification. The current database of Kraken 2 is rich in bacterial sequences but contains only few eukaryotic sequences. Therefore, nearly the whole eukaryotic diversity in our metagenomic data set was missing, while the metabarcoding data sets contained nearly only bacterial sequences.

Another major difference between both approaches was, that generally the percentage of reads classified was much higher at various taxonomic levels in the metagenomic data sets than in the metabarcoding data sets (Appendix A). For example, at the phylum level in the metagenomic and metabarcoding data sets, we classified between 70 and almost 100% and 51–89% reads, respectively. For 9 out of 16 metagenomic data sets, we obtained classification level of more than 80%. In contrast, only 6 out of 36 metabarcoding data sets reached the same percentage for taxonomic classification at the phylum level (Appendix A). We observed similar difference for the lower taxonomic levels. At the species level, the observed classification level ranges for metagenomic and metabarcoding were 7–34% and 3–24%, respectively.

Although the overall percentage of reads classified in our metagenomic data sets was low, the observed biodiversity was much higher than in the metabarcoding. For example, the calculated Shannon indices were >3.8 in the metagenomic analyses, while the values obtained in metabarcoding were much lower and in the range of 1.6 to 3.1 (with one exception; Table 1). Figure 1 compares the Shannon indices for all analyzed Antarctic and Arctic samples. The Shannon indices for the metagenomic samples (Figure 1 blue) were always higher than the Shanon indices for the metabarcoding samples (Figure 1, orange). There was only one exception. For the Lang37 site, both types of data sets showed similar Shannon indices. Lang37 was the only site where metagenomic and metabarcoding data were very similar (see Appendix A).

The difference between the Shannon indices of the metagenomic and metabarcoding data sets was due to a higher number of OTUs recognized by the Kraken 2 software in the metagenomic datasets. Many OTUs could not be found at all in the metabarcoding data sets. Even after removal of the low abundant OTUs, 10 to 60% (depending on the site and probably the number of replicates used in metagenome analyses) of the OTUs were enriched in abundance (present only or significant fold change) in the metagenomic data sets (Table 1). Only a small number of OTUs (always less than 1% of the filtered taxonomical tags) were enriched in the metabarcoding datasets. Thus, at all investigated sites, a single metagenome recovered more than 99% (often 99.9%) of the diversity as revealed by metabarcoding using up to four replicates.

In agreement with this, we observed remarkable differences between the taxa composition of the two approaches. At all taxonomic levels, the composition is clearly very different for metagenomic and metabarcoding data. As an example, Figure 2 shows the taxonomic composition for the Pad2 site at two different taxonomic levels (Figure 2A,B). At the phylum level (Figure 2A), metagenomic data sets were dominated by Acidobacteria followed by Proteobacteria and Cyanobacteria (with Acidobacteria representing nearly 60% of the classified reads).

In contrast, the metabarcoding data sets were dominated by Cyanobacteria (representing about 50% of the classified reads except for replicate 2) followed by Acidobacteria and Proteobacteria (Figure 2A). Similar results were obtained for all other sites. For all data sets we observed considerable variation between metabarcoding replicates similar to the one shown for the Pad2 site in Figure 2. These results highlight the importance of replication in metabarcoding studies.

Having established that the quantitative results were very different for metabarcoding and metagenomic analyses, we wondered whether the taxon composition at lower levels would be also very different. As we are mostly interested in phototrophs, we decided to look in more detail into the presence of cyanobacteria at the investigated sites. Table 2 shows the cyanobacterial composition at the genus level for the Pad2 site. Similar to the results reported above, metagenomics identified more genera than metabarcoding. Genera, present with a small read number in the metagenome data set, were not present at all in the metabarcoding data set. However, surprisingly, we also observed major quantitative differences between metagenomic and metabarcoding data. The cyanobacterial reads of the metagenomic data sets were dominated by *Nostoc* (approximately 26%), *Chroococcodiopsis* (approximately 6.5%), *Calothrix* (approximately 2.5%), *Scytonema* (approximately 1.5%), and *Leptolyngbya* (approximately 1.1). In contrast, *Leptolyngbya* was the dominant cyanobacterium in the metabarcoding replicates (20 to 60% in the different replicates), followed by *Nostoc,* with approximately 2% in all replicates (Table 2). Similar results were also obtained for the other sites investigated.

## 4. General Discussion

In this manuscript, we provide a comparison of metagenomic and metabarcoding approaches for studying the biodiversity of the soil environment. Both are commonly used to assess the microbial diversity in soil samples [7,19]. As we used the same DNA samples and the same bioinformatic pipeline for both molecular methods, the results should be directly comparable to each other for both approaches. The differences were surprisingly high. The bacterial biodiversity analyzed using the Kraken 2 software was much higher in all metagenomic analyses. The reason for this is mainly the detection of minor community members in metagenomic analyses which are missing in the metabarcoding data sets. This is probably due to the higher sequencing depth in metagenome studies. To allow the assembly of complete genomes for the major community members, the sequencing depth is much higher in metagenomic studies than in metabarcoding studies, which apparently benefits the detection of minor community members. This could probably also be compensated by higher sequencing depth in metabarcoding analyses.

However, a more troubling result was the big quantitative differences at every taxonomic level, e.g., Actinobacteria and Cyanobacteria (at the phylum level) and *Nostoc* and *Leptolynbya* (at the genus level). While we cannot exclude other possibilities completely, it seems most likely that the PCR amplification is causing these differences when targeting a marker gene sequence like the 16S rDNA by PCR amplification [14,19]. Different PCR amplification rates for the same gene in different species are well known, and, apparently, not only effect the composition of OTUs in metabarcoding studies at higher taxonomic levels (Acidobacteria vs. Cyanobacteria) but also at lower levels (e.g., genus: *Nostoc* vs. *Leptolynbya*).

While overall the results for different replicates for metabarcoding and metagenomics were very similar, the completely different taxon composition of for example the Pad2 replicate 2 in the metabarcoding study highlights the importance of replication. Single replicates can clearly be very misleading when analyzing the biodiversity of soil samples. However, our metagenomic analyses for of single replicate recovered more than 99% of the biodiversity observed with two to four replicates in our metabarcoding analyses, even if very different replicates were encountered in metabarcoding as for the Pad2 site. Nevertheless, the differences between both methods became even more pronounced when more replicates for both methods were included in our analyses (e.g., in Table 1, overrepresentation rates correlate with replicate number).

Finally, a major advantage of metagenomics is the ability to assemble nearly complete genomes for members of the communities. For example, a total of 321,299 reads were identified for *Nostoc* in the Metagenome 2 data set for Pad2. The reads represent 0.27% of the total reads of this data set and a total of 48 MB, which is approximately five times the size of a typical Nostoc genome (*N. punctiforme*: 8.9 MB [26]).

## 5. Conclusions

In this study, we demonstrate that metagenomic analyses are clearly better suited than metabarcoding for investigating the biodiversity of terrestrial soil samples. The number of observed species is much higher and quantitative analyses are not hindered by amplification biases in the PCR step for metabarcoding. In addition, metagenomes will allow to compare functional aspects for different communities. The only drawback of metagenome analysis is the larger data sets which requires more time for computer analyses.

## Figures and Tables

**Figure 1 genes-14-00812-f001:**
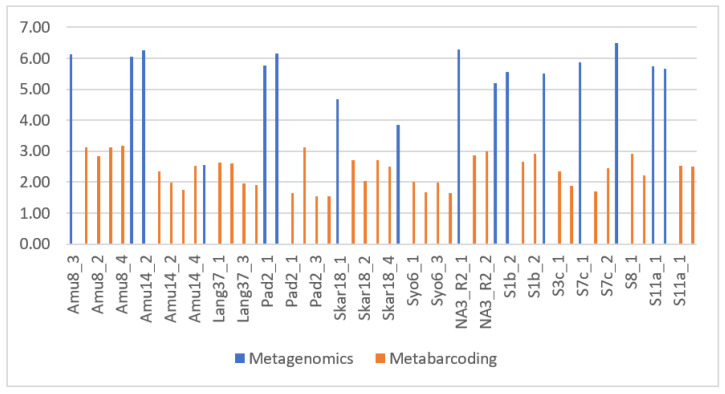
Shannon indices for all analyzed samples. Samples to the left—Amu8 to Syo6—are from East Antarctica, whereas samples to the right—NA3_R2 to S11a—are from Svalbard (the Arctic).

**Figure 2 genes-14-00812-f002:**
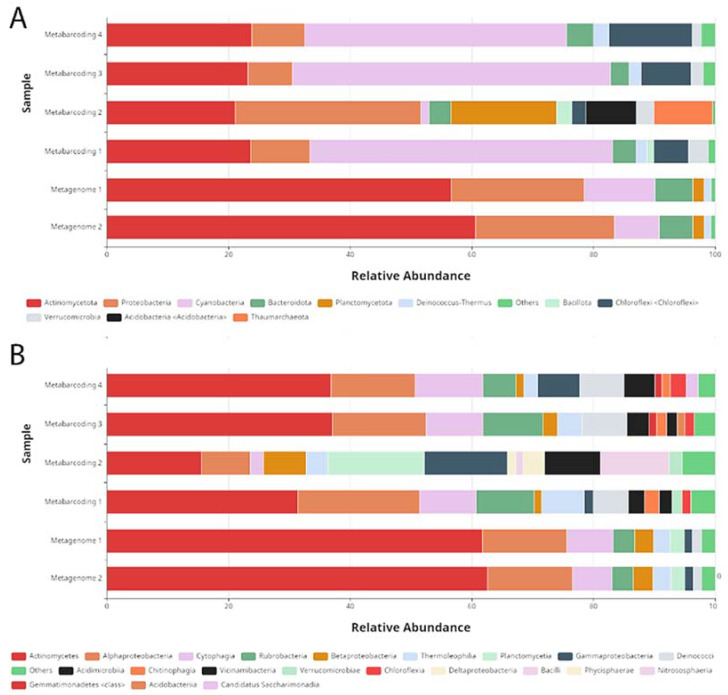
Biodiversity of the Pad2 site. The metagenomic and the metabarcoding datasets were analyzed using the Kraken 2 software and Omicsbox tools. Shown are stacked bar charts of the abundance (*x-*axis) of classified reads. For simplicity taxonomic groups with less than 1% abundance are grouped together as others. The same DNA samples (replicate 1 of the four replicates shown) were used for metabarcoding 1 and the metagenomes 1 and 2. (**A**) Phyla; (**B**) classes.

**Table 1 genes-14-00812-t001:** Differential abundance of taxonomic tags (species taxonomic level) in metagenomic and metabarcoding studies of terrestrial soil samples from polar regions. Sequences were classified with the Kraken 2 software and differential abundance analyses performed using edgeR and standard settings. The percentage of filtered taxonomic tags over- or underrepresented in metagenomes is given in brackets.

Site	Total Tags	Tags after Filtering ^(1)^	Diff. Abundant Tags	Overrepresented in Metagenomes	Underrepresented in Metagenomes
Amu8	6410	5716	1409	1402 (24.5%)	7 (0.1%)
Amu14	7055	3788	2277	2254 (59.5%)	23 (0.6%)
Lang37	5487	4014	739	734 (18.2%)	5 (0.1%)
Pad2	7482	3558	2196	2187 (61.5%)	9 (0.1%)
Skar18	5031	4496	753	737 (16.4%)	16 (0.3%)
Syo6	5342	4333	615	585 (13.5%)	30 (0.7%)
NA3_ R2^C^	n/a	n/a	n/a	n/a	n/a
S1	7103	3552	2040	2015 (56.7%)	25 (0.7%)
S3	5206	5206	509	497 (9.5%)	12 (0.2%)
S7 ^(2)^	n/a	n/a	n/a	n/a	n/a
S8	5443	5443	1681	1661 (30.5%)	20 (0.4%)
S11	6338	3585	1658	1641 (45.7%)	17 (0.5%)

^(1)^ OTUs had to be present in at least 2 samples with more than 4 reads/million reads to be included in the analysis. ^(2)^ Not enough replicates for differential abundance analysis.

**Table 2 genes-14-00812-t002:** Comparison of the abundance of selected cyanobacterial genera in metagenomes and metabarcoding analyses of the Pad2 site. Bold numbers indicate the replicate that was used for metagenomics and metabarcoding.

	Metagenome 1	Metagenome 2	Metabarcoding 1	Metabarcoding 2	Metabarcoding 3	Metabarcoding 4
Total cyanobacterial reads	965759	1279813	20809	1440	19568	14344
	Percentage of cyanobacterial reads					
*Anabaena*	**0.22**	**0.25**	**n.d**	n.d	0.01	n.d
*Calothrix*	**2.34**	**2.60**	**0.03**	0.14	0.05	0.09
*Chroococcidiopsis*	**6.42**	**6.92**	**0.11**	n.d	n.d	n.d
*Fischerella*	**0.51**	**0.57**	**n.d.**	n.d.	n.d.	n.d.
*Gloeobacter*	**0.07**	**0.12**	**0.10**	0.07	0.09	0.10
*Gloeocapsa*	**0.53**	**0.61**	**0.01**	0.14	0.01	0.01
*Gloeomargarita*	**0.02**	**0.03**	**n.d.**	n.d.	n.d.	n.d.
*Gloeothece*	**0.03**	**0.06**	**n.d.**	n.d.	n.d.	n.d.
*Leptolyngbya*	**1.09**	**1.26**	**36.27**	66.81	26.14	47.11
*Microcystis*	**0.09**	**0.10**	**n.d.**	n.d.	n.d.	n.d.
*Nostoc*	**25.90**	**25.11**	**2.03**	1.46	2.29	1.46
*Oscillatoria*	**0.20**	**0.22**	**0.02**	n.d.	0.01	0.01
*Parasynechococcus*	**p.**	**p.**	**n.d.**	n.d.	n.d.	n.d.
*Phormidium*	**0.02**	**0.03**	**n.d.**	n.d.	n.d.	n.d.
*Pseudanabaena*	**0.04**	**0.05**	**n.d.**	n.d.	n.d.	0.01
*Scytonema*	**1.38**	**1.50**	**0.09**	n.d.	0.12	n.d.
*Synechococcus*	**0.32**	**0.50**	**0.00**	n.d.	0.04	n.d.
*Synechocystis*	**0.04**	**0.05**	**n.d.**	n.d.	n.d.	n.d.
*Thermoleptolyngbya*	**0.09**	**0.11**	**n.d.**	n.d.	n.d.	n.d.
*Thermosynechococcus*	**0.05**	**0.08**	**n.d.**	n.d.	n.d.	n.d.

n.d. = not detected. p. = present.

## Data Availability

The raw reads of the metabarcoding study for the Arctic and Antarctic samples are available at the Sequence Read Archive (SRA) under projects PRJNA881983 and PRJNA936101, respectively. The raw reads for the metagenome study were submitted to SRA under project PRJNA945601 for the Eurofins data set and PRJNA948668 for the Biomarker data set.

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
