# Peer review of "Metagenomics Provides a Deeper Assessment of the Diversity of Bacterial Communities in Polar Soils Than Metabarcoding"

_genes, 2023, doi:10.3390/genes14040812_

Round 1

Reviewer 1 Report

Major concern:

The manuscript written by Becker B. and Pushkareva E. studied diversity of soil bacteria from arctic and Antarctic regions by metabarcoding and metagenomic approaches. The emergence of molecular methods based on the isolation of total microbial DNA from the soil and its subsequent analysis has become a new stage in the development of soil microbiology. Analyses of amplicons (metabarcoding) and genomes (metagenomics) are presently considered to be the predominant research approaches in soil microbiology; they provide a starting point for new research directions and hypotheses. While it is important to close the knowledge gap about polar microbes and choose the most correct method in particular, there is more work to be put into this manuscript. Additionally, the manuscript unfortunately written carelessly and should be re-checked for language errors. Some of the language issues are marked, but the list is not exhaustive.

I hope, the following comments are of use to improve the manuscript's quality.

Introduction

P.1 line 10: “Our results show; that nearly…” remove the semicolon

P.1 line 11: “In contrast; metagenomics…” remove the semicolon

P.1 line 12: “In addition; we found …” remove the semicolon

P.1 line 22: “earths” ???? May be Earth?

P. 1 lines 34, 35, 39 and others – insert space between “]” and next word and check punctuation marks.

P.2 Lines 46-48. Check the grammar.

P. 2 line 48 “a specific RNA sequence” You amplifying the rRNA gene. Please correct this.

P. 2 line 69 alrady???

M&M

The methodology needs further explaining.

P. 2 line 79 V3-4 region of rRNA change to V3-4 region of rDNA throughout the text

Projects numbers are missing.

The authors describe M&M rather briefly, referring to their earlier work, Pushkareva et al., 2022. In this study, bioinformatic analyses are done using ASV. However, later in the Results of this manuscript, they are talking about OTU. Thus, it is not clear which of the methods for determining consensus sequences was used in this work. I recommend expanding this part of M&M.

P. 3 line 108 diversity in in our metagenomics remove double in

Table 1 looks terrible. Apparently, the rows with the values of the diversity indices have been shifted. In addition, I recommend removing a few decimal places, leaving only one. They carry no meaning.

Figure 2 is unreadable. As far as I could understand, along the y-axis there are different Replicates related to one Pad2 sample. So change the name of the axis and correct Figure.

Author Response

We thank the reviewer for these thoughtful comments to improve the manuscript. We have tried to follow his/her suggestions during revision of the manuscript. Below you find a list of all changes

Major concern:

The manuscript written by Becker B. and Pushkareva E. studied diversity of soil bacteria from arctic and Antarctic regions by metabarcoding and metagenomic approaches. The emergence of molecular methods based on the isolation of total microbial DNA from the soil and its subsequent analysis has become a new stage in the development of soil microbiology. Analyses of amplicons (metabarcoding) and genomes (metagenomics) are presently considered to be the predominant research approaches in soil microbiology; they provide a starting point for new research directions and hypotheses. While it is important to close the knowledge gap about polar microbes and choose the most correct method in particular, there is more work to be put into this manuscript. Additionally, the manuscript unfortunately written carelessly and should be re-checked for language errors. Some of the language issues are marked, but the list is not exhaustive.

I hope, the following comments are of use to improve the manuscript's quality.

We address all these comment below when answering the more specific points.

Introduction

P.1 line 10: “Our results show; that nearly…” remove the semicolon

Done

P.1 line 11: “In contrast; metagenomics…” remove the semicolon

Done

P.1 line 12: “In addition; we found …” remove the semicolon

Done

P.1 line 22: “earths” ???? May be Earth?

Done

  1. 1 lines 34, 35, 39 and others – insert space between “]” and next word and check punctuation marks.

Done

P.2 Lines 46-48. Check the grammar.

Replaced with new sentences: The frequency with which an rRNA sequence is found in a sample is a measure of the abundance of an rRNA sequence. Individual rRNA sequences are commonly referred to as amplicon sequence variant (ASV) or operational taxonomical unit (OTU) and represent a species.

  1. 2 line 48 “a specific RNA sequence” You amplifying the rRNA gene. Please correct this.

Done, see above.

  1. 2 line 69 already???

Done

M&M

The methodology needs further explaining.

See below

  1. 2 line 79 V3-4 region of rRNA change to V3-4 region of rDNA throughout the text

Done

Projects numbers are missing.

The projects numbers for the metabarcoding reads are now included. The sequences for the metagenomic reads have been submitted to SRA (Submission number SUB12878568). Once we receive the project number we will add these to the manuscript.

The authors describe M&M rather briefly, referring to their earlier work, Pushkareva et al., 2022. In this study, bioinformatic analyses are done using ASV. However, later in the Results of this manuscript, they are talking about OTU. Thus, it is not clear which of the methods for determining consensus sequences was used in this work. I recommend expanding this part of M&M.

The reviewer is right, this might be misleading. In our previous metabarcoding publication with these samples, we clustered reads into ASVs with 100 % similarity. ASVs are sometimes clustered by scientists to OTUs by sequence similarity (generally 95 – 97 sequence identity). However, in this project we used the Kraken software, which works completely different and can be used on sequencing information independent of the method used, e.g. metabarcoding of amplicon sequences and metagenome sequences. This has the great advantage that all sequence information in metagenome work can be used. rRNA make only a small amount of metagenomic data, but with Kraken we can use the whole sequence information.

Briefly, Kraken is a taxonomic sequence classifier that assigns taxonomic labels to DNA short reads. It does so by examining the k-mers (generally 35-mers) within a read and querying a database with those k-mers. This database contains a mapping of every k-mer in Kraken's genomic library to the lowest common ancestor (LCA) in a taxonomic tree of all genomes that contain that k-mer. The set of LCA taxa that correspond to the k-mers in a read are then analyzed to create a single taxonomic label for the read; this label can be any of the nodes in the taxonomic tree. Kraken is designed to be rapid, sensitive, and highly precise. This approach is feasible for metagenomic WGS as well as 16S/ITS amplicon read input data. The developer of Kraken used OTUs for the taxonomic sequence classifier assigned to a read. For this reason, we use OTU also throughout the manuscript.

We have added a few sentences to the M&M section to explain the method used by Kraken.

  1. 3 line 108 diversity in in our metagenomics remove double in

Done

Table 1 looks terrible. Apparently, the rows with the values of the diversity indices have been shifted. In addition, I recommend removing a few decimal places, leaving only one. They carry no meaning.

We have moved Table 1 as Excel sheet to supplemental data, as we added more sites for our analyses (as suggested by reviewer 2) and the table got really too big.

Figure 2 is unreadable. As far as I could understand, along the y-axis there are different Replicates related to one Pad2 sample. So change the name of the axis and correct Figure.

Modified as also suggested by reviewer 2. We removed the two lower taxonomic levels to increase readability. Both levels do not add real additional value. The observed differences between different methods and replicates are found at all taxonomical levels and for all sites investigated.

In addition we have tried throughout the text to strengthen the clarity and our major findings.

Reviewer 2 Report

Comments for Authors

Manuscript ID: genes-2181817

Title: Metagenomic analyses provide a deeper assessment of the diversity of polar bacterial soil communities than metabarcoding

 Major comments:

1. The authors compared the assessment depth of bacterial community diversity within polar soils between two methods, i.e., metagenomic and metabarcoding analyses (16S rRNA sequencing), to determine which method can more comprehensively reflect the species composition of the whole bacterial communities. However, the sample number of analyses database is too limited. There are only 2 samples for metagenomic analyses and 4 samples for metabarcoding analyses, which cannot accurately reveal the assessment results. At least 3 repeated soil samples within one area and over 9 sampling areas are required. Thus, the research method should be greatly improved.

2. Generally, the innovation of this study is limited. The authors found that the taxa found in metabarcoding analyses were recovered in metagenomic analyses, while the metagenomics identified a large number of additional OTUs absent in metabarcoding analyses. This result is too common, which is absent of novelty.

 Minor comments:

Line 3: It should be “polar soil bacterial communities or “bacterial communities in polar soils.

Line 10: It should be showed that.

Line 11, 12, 14, 16: ; is misused. It should beIn contrast,, In addition,, studies,, ...... respectively. Please check these mistakes throughout the manuscript.

Line 15: What does the “minor community members” mean? Did the authors refer to the rare species/OTU (species with low abundance)?

Line 81, 85: Why did the authors provide the SRA number in forms of XXX? The authors should provide the obtained SRA number here before accepted.

Line 108: Double in? Please check it.

Line 109: It should be Data.

Line 181: The Figure 1 is too vague, which cannot be seen clearly.

Line 220: It should be In addition..

Line 221: It should be functional.

Author Response

We thank the reviewer for his/her important comments. Find below our point to point response.

    1. The authors compared the assessment depth of bacterial community diversity within polar soils between two methods, i.e., metagenomic and metabarcoding analyses (16S rRNA sequencing), to determine which method can more comprehensively reflect the species composition of the whole bacterial communities. However, the sample number of analyses database is too limited. There are only 2 samples for metagenomic analyses and 4 samples for metabarcoding analyses, which cannot accurately reveal the assessment results. At least 3 repeated soil samples within one area and over 9 sampling areas are required. Thus, the research method should be greatly improved.

We completely agree with the reviewer that replicate samples and several sampling areas are required for quantitative analyses of community diversity at different sites. However, that was not the question for this contribution to the special issue on Polar Genomics. In this communication, we do not want to solve ecological questions nor demonstrate quantitative differences between our sites. In this paper we focus on a rather simple question: Is metabarcoding or metagenomics better suited to analyse the biodiversity of microbial communities from polar soils? It is often stated that metagenomics is better, however, to our knowledge that has not been really carefully investigated using the same DNA samples and the same analysis software. We think that our analysis clearly shows that metagenomics is superior to metabarcoding and we think that this is an important information for the scientific community.

That said, we agree with the reviewer that it would be better to have more sampling sites in this analysis. Therefore, we have added 8 additional sampling sites (4 Arctic and 4 Antarctic sites). We have moved table 1, which got even bigger with the new data to the supplement. Instead we present now new Figure 1 showing the Shannon Indices for metagenomic and metabarcoding analyses. The additional results were also added to new Table 1 (former Table 2) and support our previous results. In addition we have tried throughout the manuscript to strengthen the clarity and our major findings.

    1. Generally, the innovation of this study is limited. The authors found that “the taxa found in metabarcoding analyses were recovered in metagenomic analyses, while the metagenomics identified a large number of additional OTUs absent in metabarcoding analyses”. This result is too common, which is absent of novelty.

See above for the significance of our results.

            Minor comments:

Line 3: It should be “polar soil bacterial communities” or “bacterial communities in polar soils”.

We changed the titel accordingly.

Line 10: It should be “showed that”.

Done

Line 11, 12, 14, 16: “;” is misused. It should be”In contrast,”, “In addition,”, “studies,”, ...... respectively. Please check these mistakes throughout the manuscript.

Done

Line 15: What does the “minor community members” mean? Did the authors refer to the “rare species/OTU (species with low abundance)”?

Changed to low abundance

Line 81, 85: Why did the authors provide the SRA number in forms of XXX? The authors should provide the obtained SRA number here before accepted.

The projects numbers for the metabarcoding reads are now given. The sequences for the metagenomic reads have been submitted to SRA (Submission number SUB12878568). Once we receive the project number we will add these to the manuscript.

Line 108: Double “in”? Please check it.

Done

Line 109: It should be “Data”.

Done

Line 181: The Figure 1 is too vague, which cannot be seen clearly.

Modified as suggested also by reviewer 1. We removed the two lower taxonomic levels to increase readability. Both levels do not add real additional value. The observed differences between different methods and replicates are found at all taxonomical levels and for all sites investigated.

Line 220: It should be “In addition.”.

Done

Line 221: It should be “functional”.

Done

Round 2

Reviewer 2 Report

No more comments.